# Adaptive Clustering for Point Cloud

**DOI:** 10.3390/s24030848

**Published:** 2024-01-28

**Authors:** Zitao Lin, Chuanli Kang, Siyi Wu, Xuanhao Li, Lei Cai, Dan Zhang, Shiwei Wang

**Affiliations:** 1College of Geomatics and Geoinformation, Guilin University of Technology, Guilin 541004, China; 2014012@glut.edu.cn (C.K.); 2120222033@glut.edu.cn (S.W.); 2120221996@glut.edu.cn (X.L.); 2120221968@glut.edu.cn (L.C.); 2120222044@glut.edu.cn (D.Z.); 2120222025@glut.edu.cn (S.W.); 2Key Laboratory of Spatial Information and Geomatics, Guilin University of Technology, Guilin 541004, China

**Keywords:** point cloud segmentation, point cloud clustering, large-scale point cloud

## Abstract

The point cloud segmentation method plays an important role in practical applications, such as remote sensing, mobile robots, and 3D modeling. However, there are still some limitations to the current point cloud data segmentation method when applied to large-scale scenes. Therefore, this paper proposes an adaptive clustering segmentation method. In this method, the threshold for clustering points within the point cloud is calculated using the characteristic parameters of adjacent points. After completing the preliminary segmentation of the point cloud, the segmentation results are further refined according to the standard deviation of the cluster points. Then, the cluster points whose number does not meet the conditions are further segmented, and, finally, scene point cloud data segmentation is realized. To test the superiority of this method, this study was based on point cloud data from a park in Guilin, Guangxi, China. The experimental results showed that this method is more practical and efficient than other methods, and it can effectively segment all ground objects and ground point cloud data in a scene. Compared with other segmentation methods that are easily affected by parameters, this method has strong robustness. In order to verify the universality of the method proposed in this paper, we test a public data set provided by ISPRS. The method achieves good segmentation results for multiple sample data, and it can distinguish noise points in a scene.

## 1. Introduction

With the development and popularization of laser radar technology, the use of three-dimensional point cloud processing technology has become increasingly important in urban development and construction [1,2]. There are many advantages of using point cloud data in actual production; however, there are also some limitations. Because of the large amount of data and messiness [3], it is necessary to carry out pretreatment before putting them into production. Point cloud segmentation techniques play a vital role in this process, and they can aid in understanding and perceiving complex scenes. Point cloud segmentation is intended to facilitate the extraction of geometrical and topological information from each object, and it involves dividing the point cloud data into different regions of the object and performing a fine-grained scene analysis and modeling [4,5].

Currently, the commonly used 3D point cloud segmentation methods mainly include the area-based method, model-based approach, convolutional network method, graph-theory-based approach, and edge-based method [6,7]. The region-based method can adaptively cluster adjacent points based on similarity, and this method has a good segmentation effect, but it can easily produce segmentation errors in the presence of large gaps or holes between objects [8]. The model-based approach can better fit the geometric model, and the calculation speed is relatively fast; however, this method has a poor segmentation effect for objects with complex geometric shapes and irregular structures, and it is sensitive to noise and local outliers [9]. The convolutional network method has a high learning capacity and expression capacity, and it can extract rich features from point cloud data; however, when using this method, overfitting easily occurs in the presence of a small amount of data, and a large amount of data are required for the training of the model. Furthermore, it is difficult for the model to make sense of the decision and explain the prediction of outcomes [10]. The graph-theory-based approach is based on the principle of image segmentation for segmenting point clouds, but this method lacks real-time performance [11,12]. The edge-based method is currently the most studied method [6], and it works by sensing the geometric boundary of the data point and then partitioning it into multiple independent sets of points. The goal of point cloud segmentation can be accomplished by using this method to visually detect the edges of different regions of the point cloud. The clustering method is used as a common method for detecting whether the three-dimensional point cloud in the scene belongs to the same set of points. This method is also widely used in edge detection methods. Chen et al. proposed an improved clustering method—Density-Based Spatial Clustering of Applications with Noise (DBSCAN)—and applied it to 3D point cloud boundary detection and planar segmentation, obtaining good results [13]. However, due to the clustering method, the edge detection method still suffers from some shortcomings in the segmentation of scenes.

The density-based and distance-based clustering methods are commonly employed in 3D point cloud segmentation [14]. DBSCAN is the primary representation of density-based clustering. This method achieves the goal of clustering by defining a specific domain range and density threshold, but a decision diagram needs to be drawn for its cluster center, which is amenable to manual intervention. Although some scholars have proposed solutions to this problem, they are still not perfect when applied to three-dimensional scenes, and they are vulnerable to outliers and some information loss [15]. The k-means clustering and KD-tree-based Euclidean clustering methods represent distance-based clustering methods. The k-means clustering method has good performance in most cases, but it is sensitive to the initial cluster centers, and the number of clusters needs to be specified in advance. In practice, unknown 3D point cloud scenes are not well used [16]. The KD-tree-based Euclidean clustering method can quickly search for adjacent points, and it also has a good effect on high-dimensional data due to KD-tree; however, this method usually uses a fixed cluster threshold in clustering. For different scenes, a great deal of debugging is required to obtain better segmentation results, and the use of fixed thresholds in large-scale scene applications is subject to classification errors [17,18].

To summarize, there are still some issues with the current methods in segmenting point clouds, especially in complex scenes. The edge detection method, which is the most commonly used segmentation method, also has some shortcomings due to the limitations of the cluster method. Therefore, establishing a clustering method that can accurately identify the boundaries of different objects in a scene without specifying the cluster center and adaptive clustering range is an important step to improve the segmentation accuracy of complex scene point clouds. To address the aforementioned issues, this paper proposes an adaptive clustering method for large-scale point clouds. This method adjusts the clustering threshold of each point using the characteristic parameters of the adjacent points of the point cloud to divide the points with the same characteristics into the same cluster points, and then it merges the preliminary segmented cluster points according to their standard deviation. After completing the above process, the cluster points whose number does not meet the conditions are clustered again, and scene point cloud segmentation is realized. Finally, the scattered points after segmentation are tested to determine whether they are outlier noise points. After removing the outlier noise points, the scene point cloud segmentation result is obtained.

## 2. Materials and Methods

The technical flow of the entire method is shown in Figure 1.

First, this paper adopts the local surface fitting method to estimate the normal vector of each point. Then, the density of each point of the scene point cloud is determined and sorted, and an adaptive clustering point cloud segmentation method is applied to complete the initial segmentation of the point cloud data. Next, standard deviation is introduced as the basis of cluster point merging to realize cluster point merging with the same characteristics. Based on the above segmentation results, the cluster points whose number of does not meet the conditions are proposed as scattered points for further clustering segmentation. Finally, after completing the segmentation of all cluster points, the discrete points are calibrated to determine whether they are outlier noise points.

### 2.1. Normal Vector Calculations

Existing normal vector estimation methods fall into three types: those based on the Delaunay method of triangulation, those based on the robust statistical method, and those based on the local surface fitting method [19,20,21]. In this paper, the local surface fitting method is selected. The reason for this is that, among the many normal vector estimation methods, the local surface fitting method is the most widely used, the principle is simple, and the calculation efficiency is high. The principle is as follows (Figure 2): Any point is selected from the imported point cloud data, and then m adjacent points are found using the KD-tree method. Then, *m* × 3 matrices are constructed using the m point coordinates, and the covariance matrix of the matrix is calculated (1). After completing the above process, the eigenvalues and corresponding eigenvectors are obtained according to the definition of the eigenvalues (2).
(1)$$Cov(X)=\frac{1}{m-1}{\left(X-\mu_{x}\right)}^{T}(X-\mu_{x})$$Cov(X) is the constructed matrix’s covariance matrix, *m* is the number of adjacent points searched, and $\mu_{x}$ is the average of each dimension.
(2)Cov(X)⋅x=λ⋅x⇒(λE−Cov(X))⋅x=0λ is the eigenvalue, and *x* is the eigenvector.

### 2.2. Adaptive Clustering Method

In comparison to traditional Euclidean clustering, KD-tree-based Euclidean clustering can accomplish the goal of clustering faster, but this method also has some limitations. This method can only consider scene segmentation from the distance of the point cloud, and it can only be applied to relatively simple scenes. In the presence of a large number of complex features, this clustering method is prone to classification errors [18]. In the adaptive cluster threshold clustering method proposed in this paper, the starting point is the distance, height difference, and normal vector angle between each point. The angle of the normal vector is primarily obtained based on the prior feature vector (3). Three parameters (4) are used to calculate the clustering threshold of each point with respect to the cluster center. A larger cluster threshold is used for points with the same features, and a lower cluster threshold is used for points with inconsistent features.
(3)$$\theta =\arccos(\frac{\left|a_{1}a_{2}+b_{1}b_{2}+c_{1}c_{2}\right|}{\sqrt{a_{1}^{2}+b_{1}^{2}+c_{1}^{2}}\cdot \sqrt{a_{2}^{2}+b_{2}^{2}+c_{2}^{2}}})$$$a_{1}$,$b_{1}$,$c_{1}$ and $a_{2}$,$b_{2}$,$c_{2}$ are two-point normal vector components, and θ is the angle between the two-point normal vectors.
(4)$$YC=\frac{\alpha \cdot density\cdot dis}{gc\cdot \theta }$$YC is the two-point clustering threshold, density is the average density of the point cloud data, dis is the distance between two points, gc is the height difference between two points, θ is the angle between two normal vectors, and α is the adjustment parameter.

### 2.3. Boundary Construction

In this paper, the method of finding outer contour points mainly involves the Graham algorithm (Figure 3). The detailed settlement process pseudo-code is shown in Algorithm 1.
**Algorithm 1** Pseudo-code of boundary construction.1. Initialize P = {P_1_,P_2_,...P_n_} as target cluster set
2. project all points onto a two-dimensional coordinate system
3. find the x coordinate extreme point in the projection two-dimensional coordinate system: xmin = min{xi},xmax = max{xi}, and find the y coordinate extreme point: y_min_ = min{yi},y_max_ = max{yi}, get four points: Pt_a_(x_min_,y_a_), Pt_b_(x_b_,y_max_), Pt_c_(x_max_,y_c_), Pt_d_(x_d_,y_min_); these four points are convex hull vertices.
4. Let Pt_1_ and Pt_a_, Pt_2_ and Pt_b_, Pt_3_ and Pt_c_, Pt_4_ and Pt_d_ exchange coordinates, connect four points counterclockwise, and construct the initial convex hull P_1_,P_2_,P_3_,P_4_.
5. **if** the points are in the initial convex hull **then**
6. remove the points from the point set of the outer contour to be judged
7. **end if**
8. divide the remaining point set into multiple subsets in order
9. obtain the convex hull vertices of each subset by Graham algorithm
10. use Graham algorithm to judge the outer contour of the subset to generate the final convex hull
11. Return OP-the outermost point of each cluster point


### 2.4. Cluster Merging

Due to the influence of external factors on the process of laser radar data acquisition, some clustering points may be missing, and then the point cloud that should belong to the same cluster point is divided into multiple clusters [22,23]. Therefore, it is necessary to judge the cluster points again after the initial point cloud clustering and to identify the cluster points with similar characteristics and similar clusters for cluster merging. The characteristic comparison method in this paper is to calculate the standard deviation of each cluster point elevation σ (5). σ is the most important and commonly used index to measure the degree of data variation, which can well explain the distribution of each cluster point [24,25]. The process of merging the clustering results is as follows (Figure 4): After the first classification, the cluster points are sorted according to the number of point clouds, and then the point cloud set with the first order is selected. The regular point cloud and the irregular point cloud are distinguished according to the distance and standard deviation, and then the distance of the two clustering blocks is determined according to whether the distance satisfies a specific density condition in order to determine whether it belongs to the same cluster point.
(5)$$\sigma =\sqrt{\frac{\sum_{i=1}^{n}\left({\left(h_{i}-E(H)\right)}^{2}\right)}{n}}$$n is the number of cluster point clouds, and E(H) is the average value of the cluster point elevation.

## 3. Experimentation

### 3.1. Study Data

For the experimental data in this paper, the pitch angle of a UAV was used to shoot an image of a park in Guilin, Guangxi, China, to obtain point cloud data(Figure 5). There were many buildings, trees, and features in this scene, and they had a large effect on the segmentation of the point cloud. To fully verify the segmentation effect of various point cloud segmentation methods on small, medium, and large scenes, this paper manually cuts the point cloud data from this scene to obtain original point cloud data—large-scene point cloud data, medium-scene point cloud data, and small-scene point cloud data. In the subsequent research and comparison, a variety of common segmentation methods are used to segment the three scenes to verify the segmentation effect.

### 3.2. Experimental Process

Once the corresponding experimental data were obtained, a number of adjacent points in the point cloud data were found in turn, and a matrix of coordinates in the three dimensions of m × 3 was constructed (the number of adjacent points in this paper is 8). The minimum eigenvector of the point cloud data was computed according to the method of normal vector calculus. A principal component analysis showed that the eigenvector is the normal vector of the point. The normal vector of the point was represented by the three-dimensional vector (a, b, c), and the corresponding three-dimensional vector was assigned to each point (Figure 6). By comparing the normal vectors of different areas, it can be found that the distribution of normal vectors is different for different features (Figure 7), which verifies that it is feasible to use the angle of normal vectors to segment features according to different features.

After obtaining the normal vector of each point, it is necessary to determine and sort the density of each point in the point cloud data. At the same time, the nearest point to the point is found, and its distance is calculated. After adding the distance between the two points to the sum of the distances, the average density of the scene is evaluated. Sorting the density of point clouds in the study area allows for high-density points to be preferentially selected as the cluster center in the subsequent clustering method. Based on the distribution of high-density points in the figure (Figure 8), the high-density points can be found to be primarily continuous points, such as buildings. Sorting the point cloud data based on density allows for the preferential extraction of regular surfaces, which improves segmentation accuracy.

At the end of the point cloud density sorting, the first point in the original point cloud data was selected as the center of the cluster. KD-tree was used to find points near the certain cluster center density relation (this paper conducted the search within four times the average density range), computed the cluster thresholds corresponding to those points and the cluster center, and determined whether the distance from these points to the cluster center was within the threshold range. Points within the threshold range were classified as belonging to the same cluster. After completing the search point test within the cluster range, it was determined whether there were any new points. If new points were found, the outermost new points were chosen as a new cluster center; if no new points were found, this clustering was halted, and the clustering results were output. Then, the repetition of the above process was eliminated to obtain multiple cluster points. It was found that the scene was divided into multiple cluster points (Figure 9), and some of the connected points clusters were also divided into multiple clusters. This is mainly due to the fact that, in order to reduce computational complexity and improve computational efficiency, once the search points were classified into the same clusters, the newly added points were not used as the centers of the clusters one by one, and the neighboring points of the newly added points were searched to determine whether they were the same cluster points. Compared with this method, the method proposed in this paper uses the farthest point from the cluster center point in the newly added point as the cluster center, and it searches for its adjacent points. Therefore, this method will inevitably cause the point cloud set that should belong to the same cluster point to be divided into multiple cluster points; however, in the following text, fine clustering and merging are carried out to solve this problem, and this method also avoids multiple operations, which strengthens its operation efficiency.

After the initial clustering segmentation was completed, the cluster points insufficient in number were eliminated (Figure 10a). According to the segmentation results, the outer contour was generated (Figure 10b), and it was determined whether it belonged to the same cluster point based on the standard deviation. After that, clusters whose elevation and standard deviation meet a certain threshold were merged (Figure 10c). After completing the above process, the regular feature segmentation was completed (Figure 10d). 

After the initial merging clustering results were obtained, the point cloud blocks with a number of scattered points and cluster point clouds of less than 20 were extracted (Figure 11). It was found that some of the extracted points were located at the boundary of the cluster block. The reason for this is that the previous normal vector calculation method was skewed at the boundary angle; the red point cloud contained a small number of cluster point clouds, and these cluster point clouds were mostly scattered forest points or small area features. These low-density clustering blocks were removed to obtain the target clustering point cloud (Figure 12a), and the corresponding two-dimensional bounding box was constructed (Figure 12b).

The above process removes most of the ground objects and avoids the interference of similar ground objects. The segmentation of ground objects can be completed using Euclidean clustering. This paper used a 3 m clustering threshold to supplement the clustering of low-density point clouds (Figure 13a). After clustering the low-density point cloud, the corresponding two-dimensional boundary was constructed (Figure 13b). Given that the distribution of point clouds in certain regions where dense forest points being inevitably encountered in the previous clustering process is relatively regular, it is necessary to judge the newly obtained clustering point cloud and the previous clustering point cloud according to the previous point cloud dispersion and the nearest distance, and the merger is expected to belong to the same cluster feature (Figure 14). Once the aforementioned work is completed, scene point cloud segmentation is essentially complete, but there are still a few scatter points that are not classified into clusters (Figure 15). Thus, in the final step, the discrete points must be re-calibrated, and the distance is used to determine whether the scattered points belong to similar blocks of cluster point clouds. The point is incorporated into the cluster point cloud if the distance satisfies a certain threshold, and, ultimately, the result of scene point cloud segmentation is achieved (Figure 16).

### 3.3. Analysis of Our Method

The purpose of the segmentation presented in this paper is to intercept the average scene from a small scene and the full scene from a large scene. In the three large, medium, and small scenes, the numbers of over-segmentation and under-segmentation point clouds of the method in this paper are compared to those of other node segmentation methods. The clustering of scene segmentation was observed based on manual segmentation results, and the superiority of the algorithm was comprehensively assessed. In this paper, we used the Euclidean clustering method with multiple cluster thresholds, the region-growing method, and the proposed method for scene clustering and segmentation to obtain segmentation results for a comparison of the point cloud data of the small scene, the medium scene, and the large scene.

The Euclidean clustering method was used with three thresholds to test the segmentation capability of small scenes. It was found that, while the 2 m cluster threshold method is able to separate the building point cloud from the ground (Figure 17), trees cannot be merged into the same cluster, and the same building belongs to different clusters. Even though the 2.5 m clustering threshold is used to ameliorate the tree clustering problem, the same cluster is also split into the same cluster. When the result of the decomposition is still poor, the 3 m cluster threshold can be used to merge trees into the same cluster. In the clustering process, the remaining ground points are merged. The region-growing method does not suffer from the aforementioned problem. The segmentation results showed that it is not possible to complete the segmentation goal for a region with a large change in the tree corner threshold (Figure 18) or determine whether there is a partial angle between two points that are expected to belong to the same cluster. In addition, the region will be split into multiple clusters. Compared to the previous method, the proposed method is significantly improved in terms of the segmentation of small scenes (Figure 19). Segmentation is performed on different patches of the same building, and the discrete points of the scene can also be extracted. Some of the points shown in the plot are not included in the clusters. In the sparse results, two cluster points can be found to be far apart (Figure 20), and in an analysis of the full scene, it is found that the points in this region are in fact the points of other trees in the scene cutting process, which must be defined as noise points. However, due to the large number of point clouds, they are redefined as a cluster. To summarize, to achieve the goal of segmenting small scene point clouds, the method proposed in this paper is the most efficient method, and the number of segmented clusters is also the closest to the true number. In fact, the clustering error of partial partition is the clipping process error, which has little to do with the method proposed in this paper.

The Euclidean clustering method was also used with three thresholds to test the segmentation ability of the medium scene. The results showed that the 1.5 m clustering threshold has a good effect on building patches (Figure 21); however, in ground point segmentation, some areas with interval holes are divided into multiple clusters, and the flowerbed point cloud of most ground buildings is also divided into ground points. The 2 m clustering threshold is used to improve the problem in which some ground points are divided into multiple clusters, but it also merges similar patches of the building, and it does not have a good segmentation effect on the near-ground flowerbed problem. The 3 m clustering threshold is only used to divide the far-distance point cloud clusters into multiple different clusters, and the segmentation effect is not observed in similar regions. Compared with the Euclidean clustering method, the region-growing method can achieve the segmentation of perigee flowerbeds (Figure 22), but it cannot achieve a good effect for discrete forest points or partial building patches. Compared with the first two methods, the proposed method achieves good results in scene segmentation (Figure 16), and the clusters obtained by segmentation are also similar to the actual ones.

The Euclidean clustering method was also used with three thresholds to test the segmentation ability of the large scene. The results showed that the 1.5 m clustering threshold (Figure 23) has a good effect on the segmentation of building patches. However, the dense forest, which has more scattered points, is divided into multiple clusters. The 2 m clustering threshold is used to improve the problem in which some ground points are divided into multiple clusters. It also merges the similar patches of the building, and it cannot achieve a good segmentation effect for the near-ground flowerbed problem. The 2.5 m clustering threshold is only used to divide the point cloud clusters that are far away in multiple different clusters. The segmentation effect is not observed in similar regions; in contrast, the region-growing method has a good segmentation effect on the plane building surface (Figure 24), but it still cannot complete the segmentation of the dense forest and other areas. In addition, it cannot achieve a good segmentation effect in some areas with slow angle changes. As for our method, the segmentation results of the large scene demonstrate good accuracy (Figure 25), and it can well separate the various features in the scene. Compared with the above methods, the number of clusters in the segmentation results is also the closest to the actual results.

In order to further verify the superiority of the proposed method, this paper selected some areas in the scene (Figure 26 and Figure 27). Because the segmentation effect of the comparison methods was not good in the dense forest area, the building and ground point cloud was chosen as the comparison area, and the comparison area was manually segmented. Based on the manual segmentation, the numbers of under-segmented and over-segmented point clouds of the three methods with regard to the cluster were counted. It can also be seen in the segmentation diagram of each sample that, in the local area, the proposed method is more stable and accurate than the other methods (Figure 28, Figure 29 and Figure 30). The numbers of under-segmented and over-segmented point clouds in each sample are calculated, and a relevant table is composed (Table 1). According to the table, error statistics are determined (Figure 31). It can be seen in the result radar map that the method proposed in this paper is superior to the other methods for each sample area; however, its error rate for sample 4 is slightly higher than that of the Euclidean clustering method. The reason for this is that manual segmentation will inevitably produce some errors. Additionally, the sample 4 area is far from the other areas in the scene; thus, it can be easily segmented using Euclidean clustering. However, in general, the segmentation method proposed in this paper is more stable and has a higher accuracy.

In order to verify the robustness of the proposed method, this paper used different clustering threshold constants for the small- and medium-sized scenes. The experimental results show that the use of different constants in small scenes does not affect simple scene segmentation (Figure 32). The segmentation effect is less impacted by the constant change, and the scene segmentation cluster does not fluctuate. For medium-sized scenes, the segmentation effect of various objects in the scene does not fluctuate greatly (Figure 33). Except for a small part of the region that increases the constant coefficient, the segmentation feature part of the point cloud is merged by other clusters, but this area is not distributed in the entire scene, and the number of clusters after scene segmentation does not fluctuate significantly. Through the change of two sets of scene parameter coefficients, it can be verified that the method in this paper has strong robustness. Compared with Euclidean clustering, where changing parameters will produce different segmentation results, the change of parameter coefficients in a certain range will not have a great impact on the final results.

In order to verify the universality of the method proposed in this paper for various sample scenarios, this paper selected a public data set provided by ISPRS for verification, and a public data set with continuous ground point cloud block data was selected for clustering. Based on the number of continuous ground point clouds in the data set and the actual number of clustered ground point clouds, the accuracy of the clustering results of the proposed method was verified (Figure 34). In the segmentation results, it can be seen that the clustering method proposed in this paper can effectively segment the ground points of multiple samples. However, there are a large number of missing segmentation points in the segmentation results, and the reason for this phenomenon is that there is a discontinuous phenomenon in some areas of the ground points in the scene, and the clustering results belonging to the same cluster cannot be effectively merged. There are over-segmentation points in some scenes. These points are mostly too close to the ground area and continuous points; therefore, these over-segmentation points cannot be effectively separated from the ground points.

In order to further reflect the advantages of this method over other methods, this paper selected a variety of methods to segment the above samples and obtained error data to make a corresponding error table (Table 2, Table 3, Table 4, Table 5, Table 6 and Table 7). In the results table, it can be seen that the proposed method is more stable than the other clustering segmentation methods. Compared with the Euclidean clustering method using multiple thresholds, the proposed method can adaptively adjust the threshold and obtain better segmentation results. In some samples, the proposed method is compared with the combination of RANSAC and the region-growing method. The results are poor, but the comparison is not large, and the ground points in these sample areas are mostly flat areas. This method cannot achieve better segmentation results in the face of images with large height differences and distances. Based on all the above results, the proposed method is more stable than the other methods and can be applied to a variety of sample areas. The clustering segmentation results can also achieve good results. The method proposed in this paper accounts for a large number of over-segmentation points, but the reason for this phenomenon is mainly due to the discontinuity of the ground area. After removing the discontinuous ground points, the number of over-segmentation points is reduced (Table 8).

## 4. Discussion

(1)How can adaptive clustering segmentation be realized?

Based on the experimental results above, it can be seen that, compared to other point cloud clustering segmentation methods, our method can more accurately cluster point clouds with certain spatial characteristics in a scene. When conducting experimental tests on public sample data sets, this method can guarantee accuracy in the case of clustering as many ground points as possible. Although the clustering effect is not good in the face of very close areas, all current clustering algorithms cannot achieve good segmentation results for this situation. Once the planar rough patch is classified, based on the scatter of the cluster points, the question of whether there is a correlation between each small cluster point is analyzed. Cluster points are merged if both the features are similar and the distances are similar. The standard deviation between cluster points is used in the merging process as a basis for judging whether the features of the cluster points are similar. Standard deviation is also used as a means of measuring the scatter of point clouds. By analyzing the discrete situation between point clouds, the overall trend of the vertical direction of the point cloud clusters is constrained, and then it is determined whether two points belong to the same cluster as a function of distance, which can allow for a better analysis of cluster points from two plane and vertical aspects. The experimental results show that our method can also realize the combination of different cluster points. In the workflow of this paper, high-density or angle-like point clouds are first combined, and then the combination of scattered point clouds is complemented. The reason for this is that, in the point cloud data, ground or building points tend to be regular, and the angle change relation is slow. The above process can be used to complete the separation of ground points. Typically, these regular point clouds occupy more than half of the scene. If we first remove these large-area point clouds and then judge the cluster of discrete point clouds, it is possible to avoid the interference of large-area point clouds and to effectively improve the accuracy and efficiency of point cloud data segmentation. For these sparse point clouds, good segmentation accuracy can be obtained using the traditional Euclidean clustering method. Once these tasks are completed, because the normal vector of the bounding region is subject to normal vector deflection relative to the central portion in the judgement process of the point cloud normal vector, discrete points must be judged after the final clustering is completed. This is based primarily on whether the distance between the discrete points and the closest points in the cluster satisfies a certain threshold, and then it can be determined whether these discrete points belong to the cluster or whether they are noise points generated during the point cloud data acquisition process.

(2)Where can the proposed method be used?

The method proposed in this paper is based on the spatial characteristics of the objects in a point cloud scene, with the aim of achieving point cloud clustering segmentation. Through this method, the extraction of objects can be effectively realized, and the segmentation results in the experimental comparison process show that the method has a smaller error than manual segmentation. Through this method, scene objects are finely modeled, the labor cost is reduced in the process of point cloud data extraction of scene objects, and the efficiency of model construction is improved.

(3)What are the shortcomings?

There are also some shortcomings to the method in this article. The process of judging the discrete point cloud is based on the traditional European clustering method, but it can better complement the sparse point segmentation of dense forest points. In the dense forest segmentation process, the points in the dense forest are merged in the same cluster, failing to achieve the goal of good single-plant separation. This will be the focus of follow-up research. In addition, there will be a deflection in the normal vector angle of the boundary point cloud in the process of judging the normal vector, and this leads to the need to make a discrete point judgment in the subsequent computational process, which reduces the efficiency of the proposed method’s operation.

## 5. Conclusions

With the goal of solving the point cloud segmentation problem in current urban scene segmentation, in this paper, we propose an adaptive clustering method based on the point cloud normal vector and sparsity. By extracting clusters of regular point clouds for different features of each point in a scene, sparse point clouds are classified. The proposed method is compared to the European clustering method and the region-growing method, commonly used in urban scene segmentation. This paper is based on manual segmentation, and the final results show that the segmentation results of our method are significantly improved compared to those of the two methods mentioned above. By comparing the segmentation clusters, it can be seen that the number of clusters segmented by our method is nearly the same as that segmented manually. To test our method, the final robustness test conducted in this paper uses different constant coefficients. This test demonstrates that, while different constant coefficients will alter the number of partial point clouds of clusters after segmentation, they have no effect on the number of clusters being segmented. There is not much difference in the segmentation results when using different coefficients, indicating strong robustness. In order to verify the efficiency of the proposed method, we calculated the speed of point cloud clustering. We used an Intel Core i3-8350K @ 4.00 GHz quad-core processor, 32 GB (Kingston DDR4 3200 MHZ 32 GB), NVIDIA Quadro P1000 graphics card, and Asus PRIME-Z370-A (Z370 chipset) motherboard to configure a computer to run the point cloud clustering program for different samples. The average processing time of one point was 0.000721 s.

This article will be strengthened in the following aspects: (1)Although the overall segmentation effect of the method proposed in this paper is good, the normal vector of the point cloud is calculated using the adjacent point in the calculation process; therefore, the normal vector of the boundary part of the scene object has a large deflection, and it is easy to eliminate the boundary points from the clustering in the calculation process of the clustering threshold.(2)In the adaptive clustering method proposed in this paper, only a simple formula for computing positive and negative ratios is used. We plan to fit and verify parameters from multiple test areas and use ablation studies to evaluate the impact of key components and then compute the relation.

## Figures and Tables

**Figure 1 sensors-24-00848-f001:**
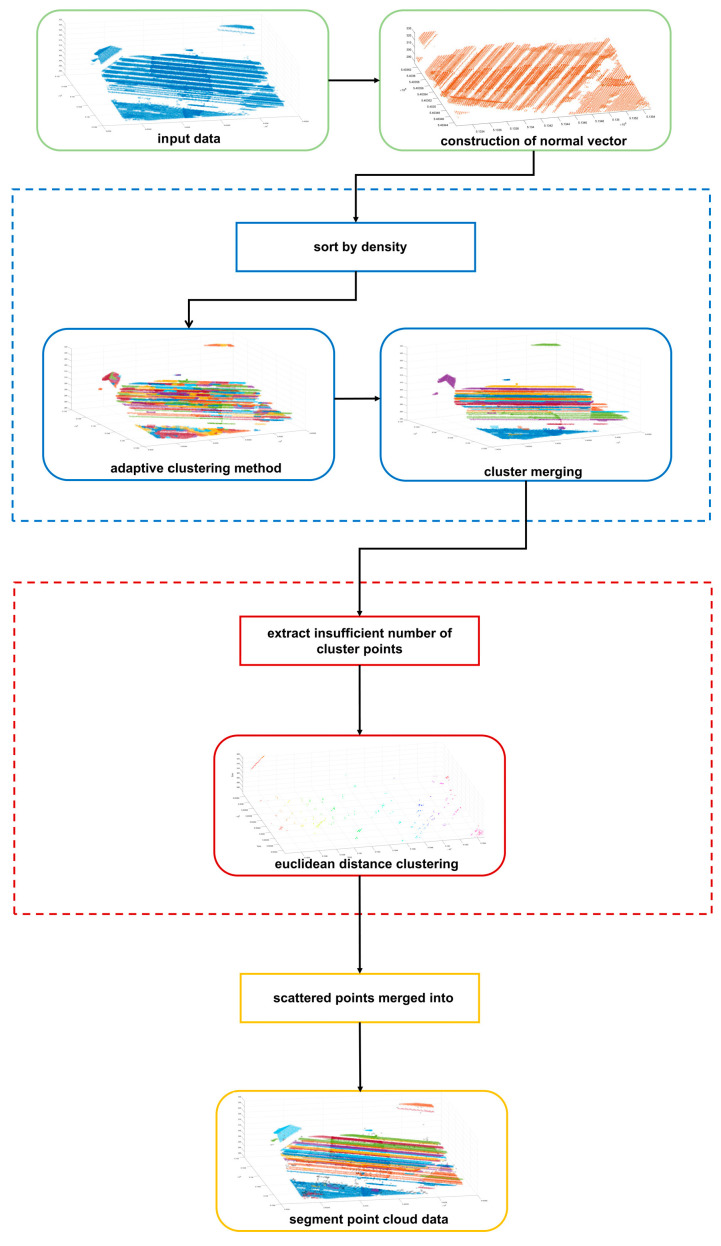
The workflow of large-scale scene segmentation based on the adaptive clustering method. Sample 42 provided by ISPRS is used as an example.

**Figure 2 sensors-24-00848-f002:**
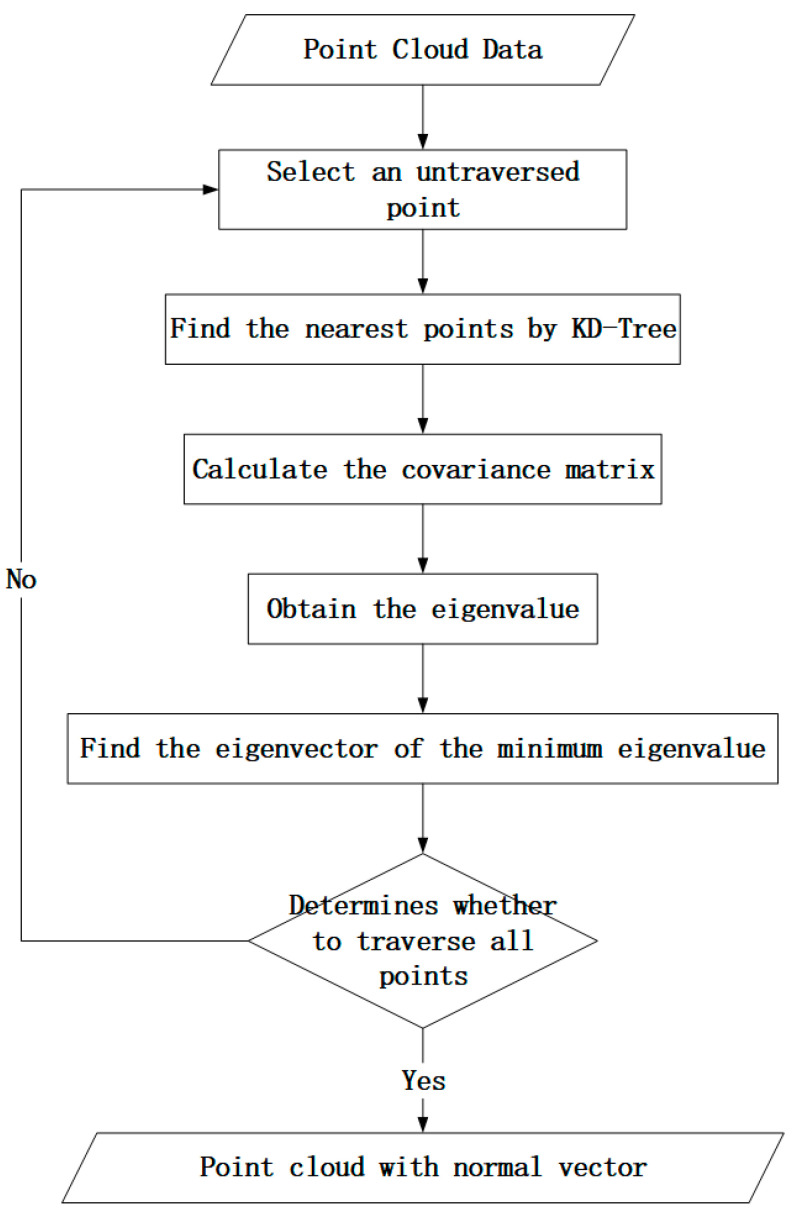
Flowchart of normal vector calculation.

**Figure 3 sensors-24-00848-f003:**
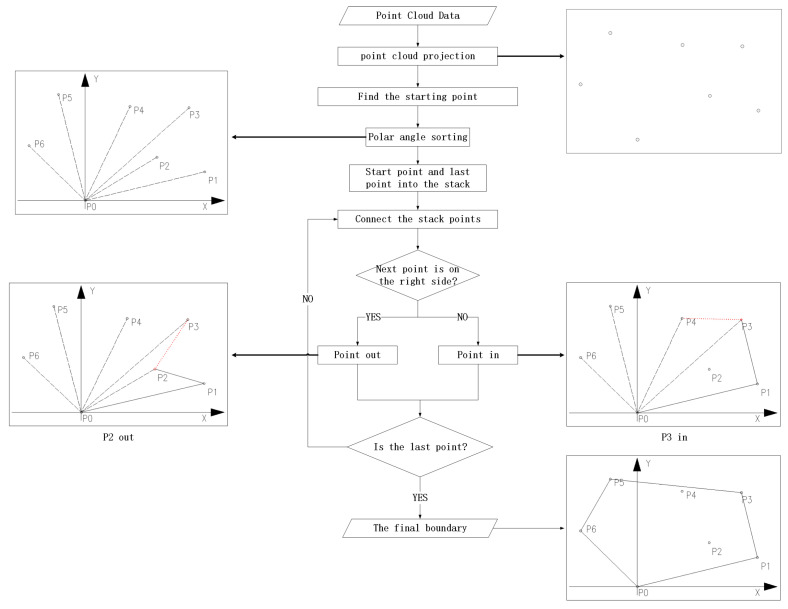
Flowchart of computation bound.

**Figure 4 sensors-24-00848-f004:**
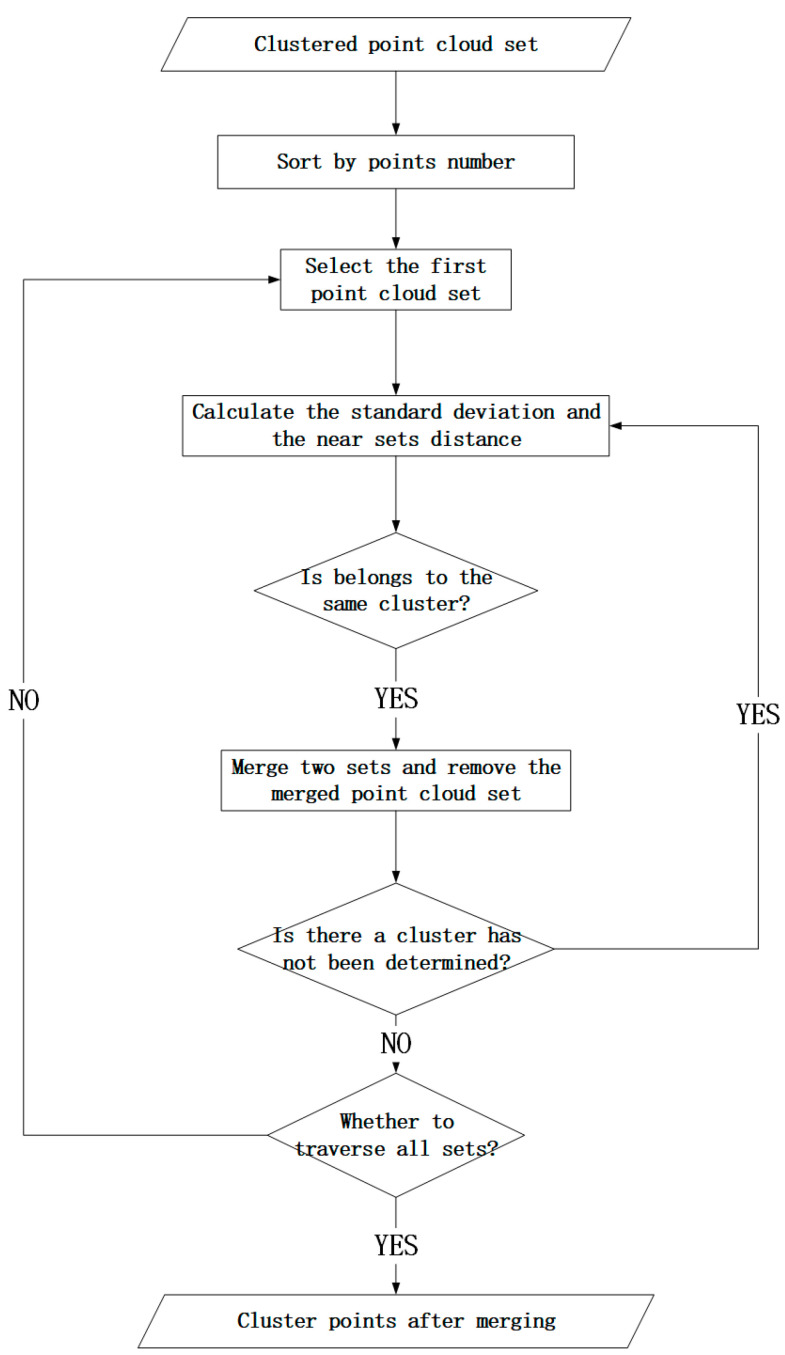
Flowchart of cluster merging.

**Figure 5 sensors-24-00848-f005:**
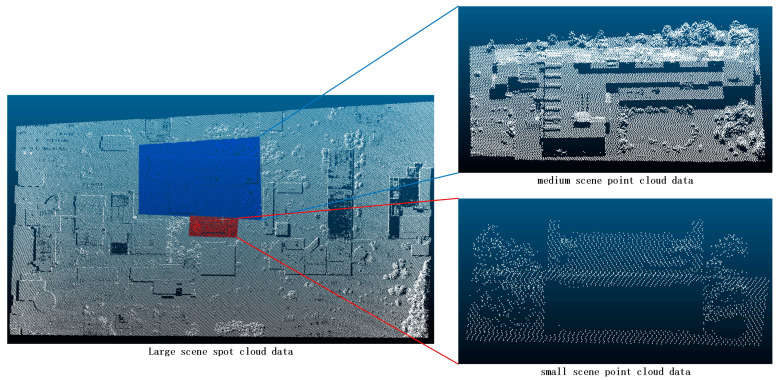
Study data distribution. White points are the original large scene study area, blue points are the medium scene study area, and red points are the small scene study area.

**Figure 6 sensors-24-00848-f006:**
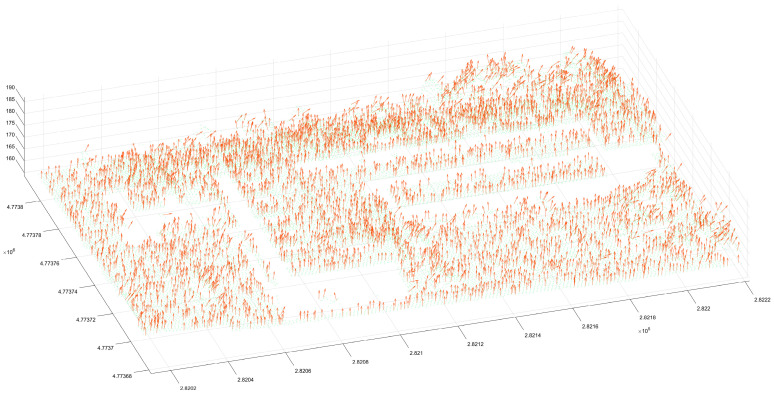
The normal vector pointing of the whole study data. A normal vector is drawn at an interval of 6 points.

**Figure 7 sensors-24-00848-f007:**
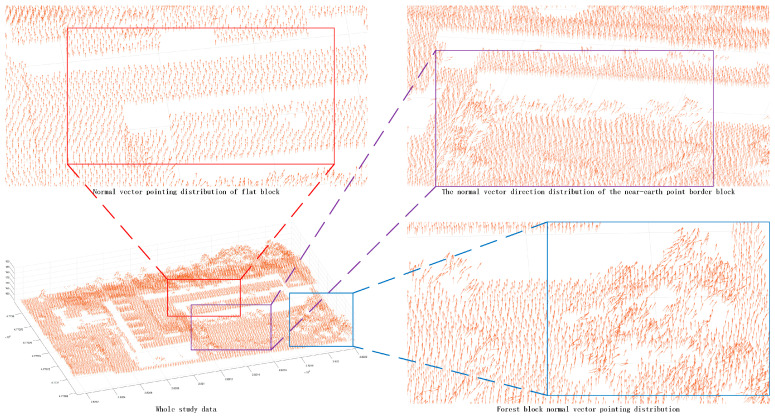
The normal vector pointing distribution of each block. The purple area is the border area, the blue area is the forest area, and the red area is the regular area.

**Figure 8 sensors-24-00848-f008:**
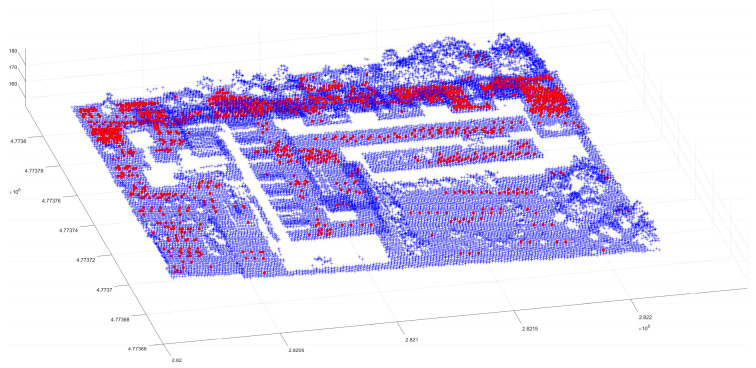
High-density point distribution. The red points comprise more than 30 adjacent points within 4 times the average density, and the blue points comprise the other points in the scene, except for the high-density points.

**Figure 9 sensors-24-00848-f009:**
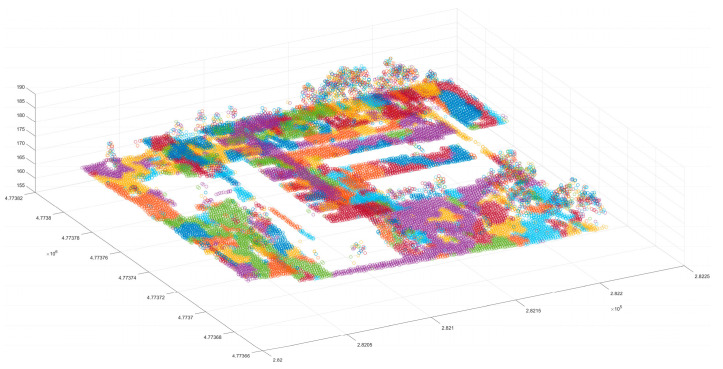
Initial clustering results. Different colors represent different clusters.

**Figure 10 sensors-24-00848-f010:**
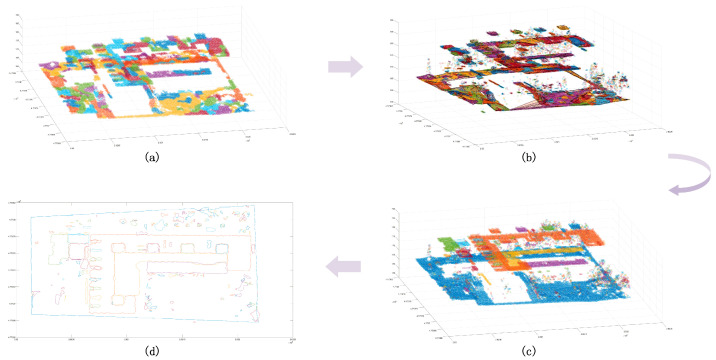
Classification results of point clouds. Panel (**a**) is the initial classification results of a certain number of point clouds, and different colors represent different clusters; panel (**b**) is the three-dimensional boundary of preliminary clusters, and the red bounding box indicates the same cluster class; panel (**c**) is the classification results with scatter points; panel (**d**) is the boundary distribution of clusters.

**Figure 11 sensors-24-00848-f011:**
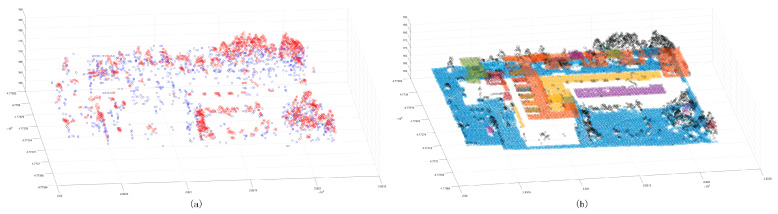
The result of low-density clustering points and merged clustering points. Panel (**a**) shows the distribution of elimination points; the blue points are scattered points, and the red points are points with a certain density. Panel (**b**) shows the distribution of segmentation results; the black points are elimination points.

**Figure 12 sensors-24-00848-f012:**
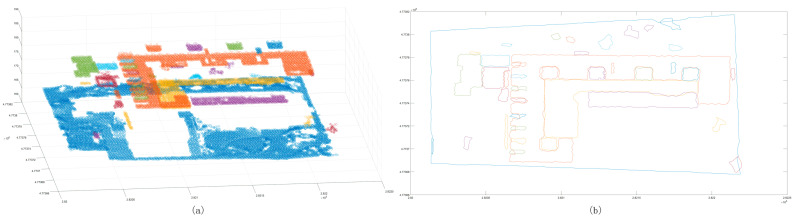
The clustering results after removing the low-density clustering point cloud blocks. Panel (**a**) shows the distribution of the clustering points after removing the low-density clustering point cloud blocks; panel (**b**) shows the boundary after removing the low-density clustering point cloud.

**Figure 13 sensors-24-00848-f013:**
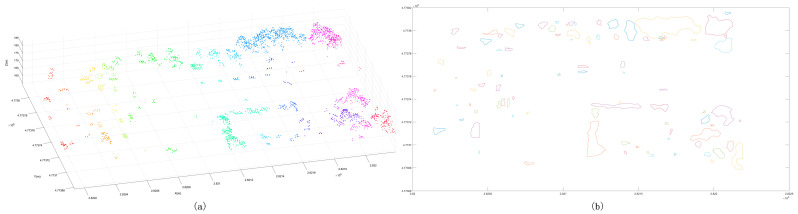
Low-density clustering results. Panel (**a**) shows the low-density point cloud clustering results; panel (**b**) shows the low-density point cloud clustering boundary.

**Figure 14 sensors-24-00848-f014:**
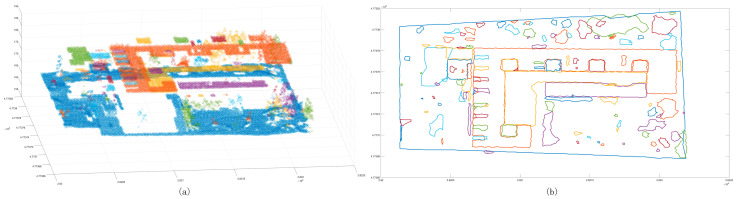
The merged clustering point cloud and its boundary. Panel (**a**) shows the distribution of clustering points, and panel (**b**) shows the boundary of the clustering points.

**Figure 15 sensors-24-00848-f015:**
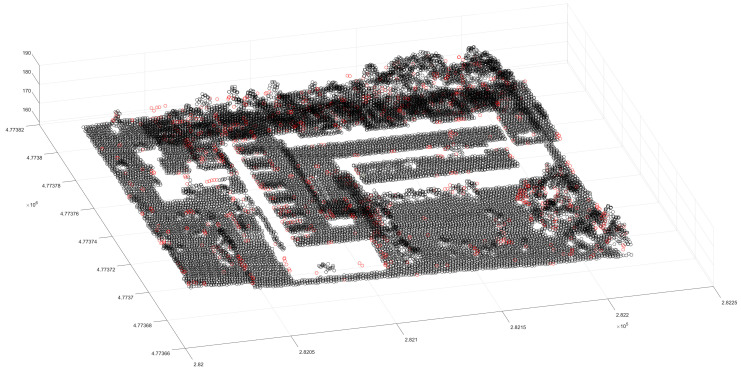
The remaining scatter distribution. The black points are the completed cluster segmentation points, and the red points are the cluster scatter points.

**Figure 16 sensors-24-00848-f016:**
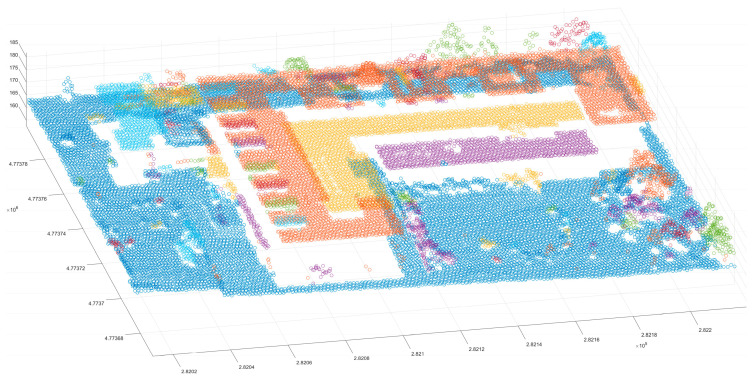
The final segmentation result.

**Figure 17 sensors-24-00848-f017:**
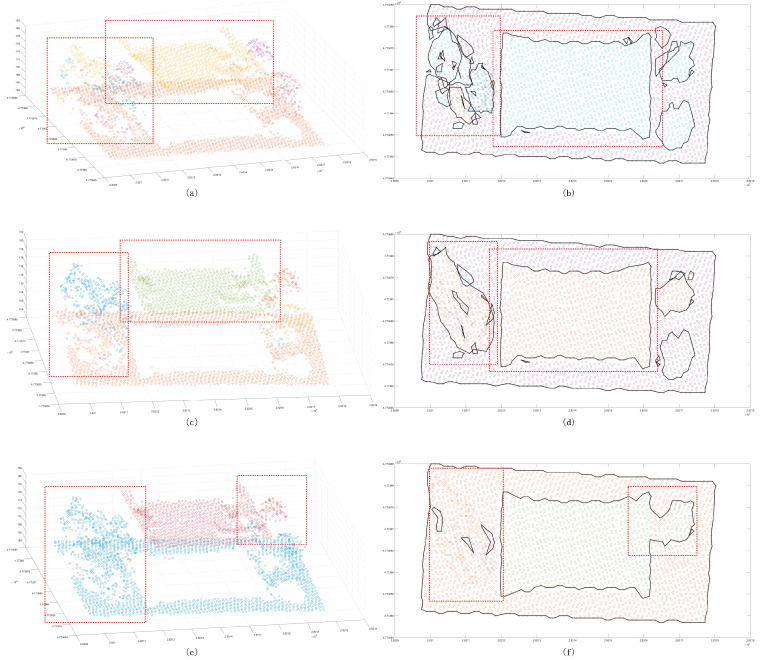
Euclidean clustering segmentation results for the small scene. Panels (**a**,**c**,**e**) show scatter distribution maps using 2 m, 2.5 m, and 3 m clustering thresholds, respectively; panels (**b**,**d**,**f**) show the cluster top views using 2 m, 2.5 m, and 3 m clustering thresholds. The black line shows the outer contour of the clustering result, and the red line box shows the area with a poor classification result.

**Figure 18 sensors-24-00848-f018:**
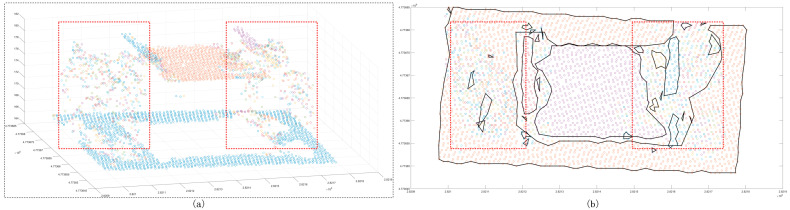
Region growth segmentation results for the small scene. The red box shows the area with a poor classification result. Panel (**a**) shows the distribution of clustering points, and panel (**b**) shows the boundary of the clustering points.

**Figure 19 sensors-24-00848-f019:**
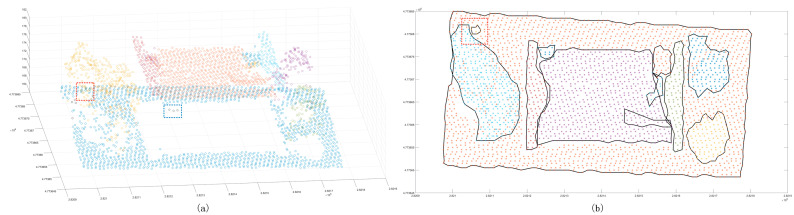
Segmentation result map of the proposed method for the small scene. The red box shows the area with a poor classification result, the blue box shows the eliminated discrete points, and the black line is the outer contour of the clustering result. Panel (**a**) shows the distribution of clustering points, and panel (**b**) shows the boundary of the clustering points.

**Figure 20 sensors-24-00848-f020:**
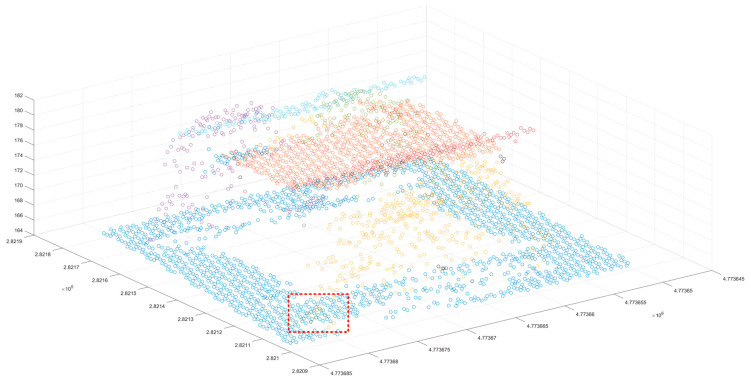
Segmentation side view of the proposed method for the small scene. The red box shows the disputed area.

**Figure 21 sensors-24-00848-f021:**
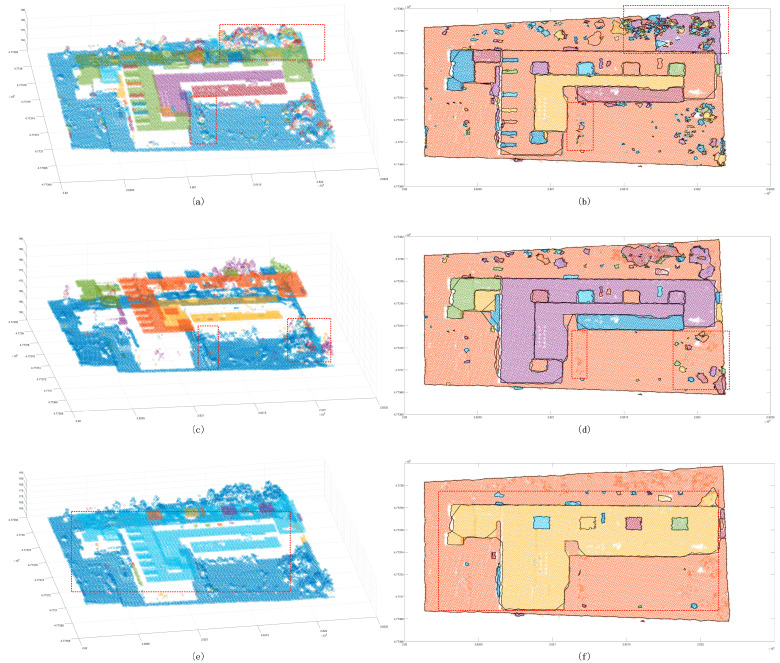
Euclidean clustering segmentation results for the medium scene. Panels (**a**,**c**,**e**) show the scatter distribution maps using 1.5 m, 2 m, and 3 m clustering thresholds, respectively; panels (**b**,**d**,**f**) show the cluster top views using 1.5 m, 2 m, and 3 m clustering thresholds. The black line is the outer contour of the clustering result, and the red line box shows the area with a poor classification result.

**Figure 22 sensors-24-00848-f022:**
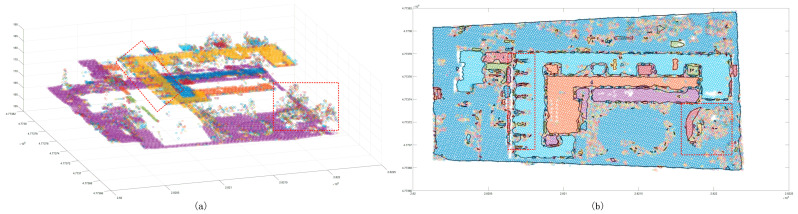
Region-growing method segmentation results for the medium scene. Panel (**a**) shows a scatter plot, and panel (**b**) shows a cluster top view. The black line is the boundary of the clustering result, and the red box indicates the area with a poor classification result.

**Figure 23 sensors-24-00848-f023:**
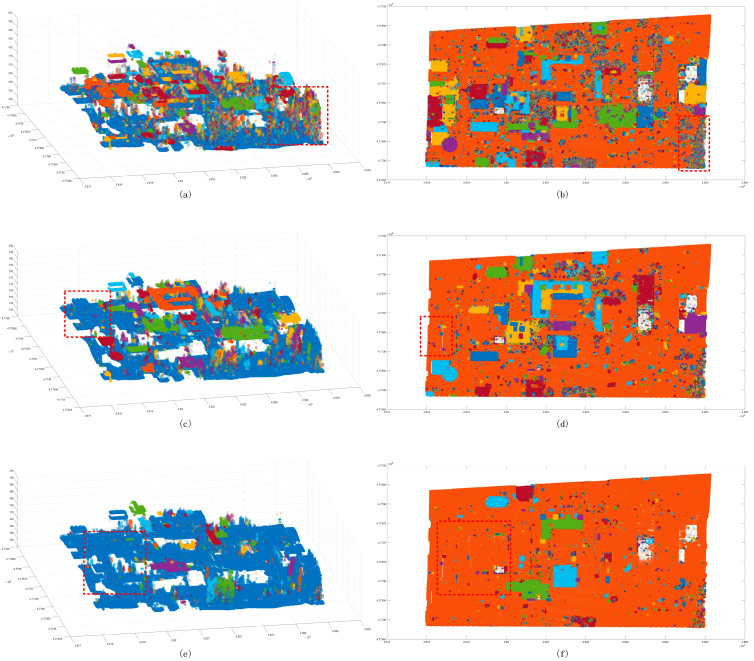
Euclidean clustering segmentation results for the large scene. Panels (**a**,**c**,**e**) show the scatter distribution maps using 1.5 m, 2 m, and 2.5 m clustering thresholds, respectively; panels (**b**,**d**,**f**) show the cluster top views using 1.5 m, 2 m, and 2.5 m clustering thresholds. The black line is the outer contour of the clustering result, and the red line box shows the area with a poor classification result.

**Figure 24 sensors-24-00848-f024:**
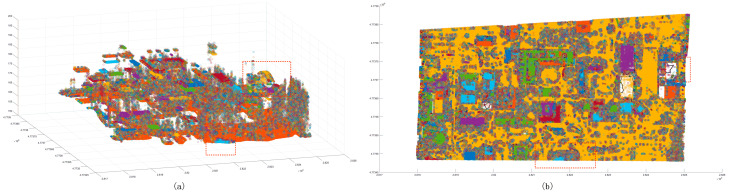
Region-growing segmentation results for the large scene. Panel (**a**) shows a scatter plot, and panel (**b**) shows a cluster top view. The black line is the boundary of the clustering result, and the red box indicates the area with a poor classification result.

**Figure 25 sensors-24-00848-f025:**
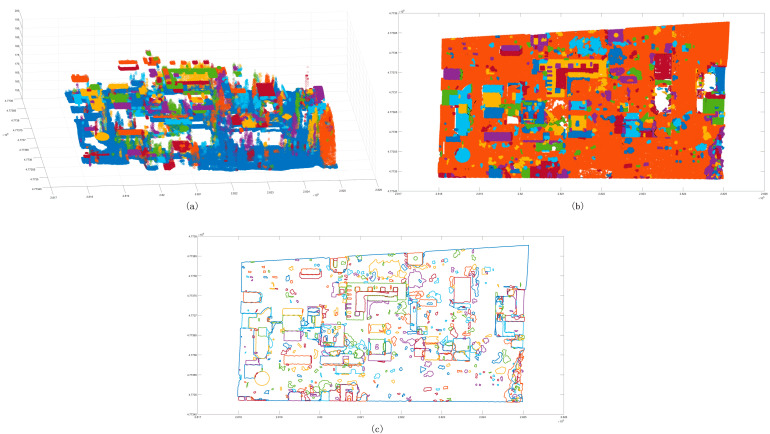
Segmentation result map of the proposed method for the large scene. Panel (**a**) shows the clustering plots, and panel (**b**) shows a cluster top view. The black line is the boundary of the clustering result. Panel (**c**) is the boundary of the clustering result.

**Figure 26 sensors-24-00848-f026:**
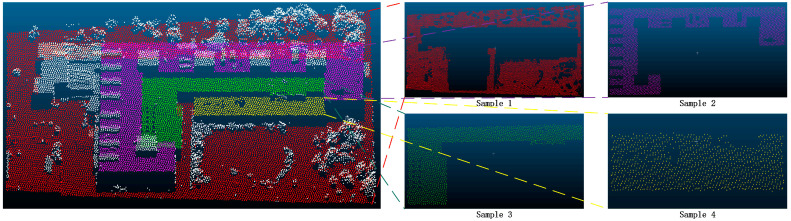
Distributions of samples 1, 2, 3, and 4.

**Figure 27 sensors-24-00848-f027:**
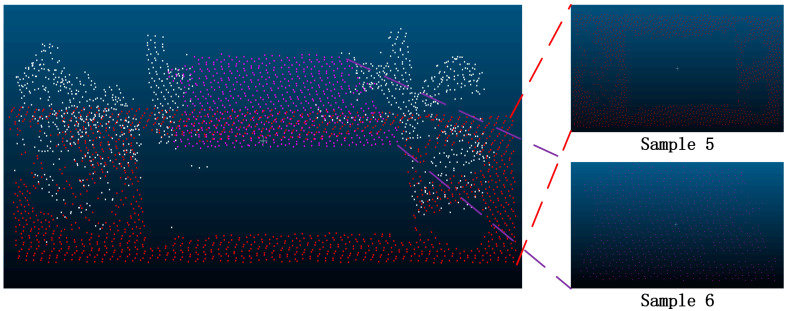
Distributions of samples 5 and 6.

**Figure 28 sensors-24-00848-f028:**
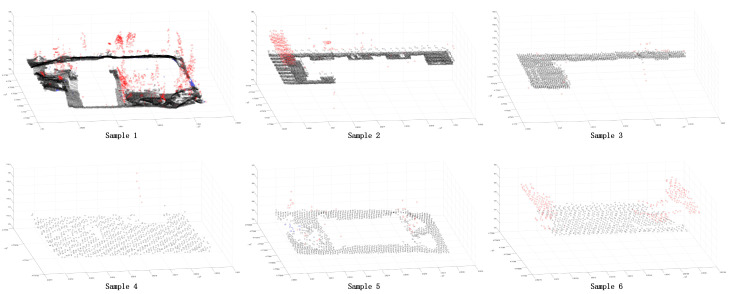
Segmentation error of each sample area using the Euclidean clustering method. The black point is the correct segmented point cloud, the red point is the under-segmented point cloud, and the blue point is the over-segmented point cloud.

**Figure 29 sensors-24-00848-f029:**
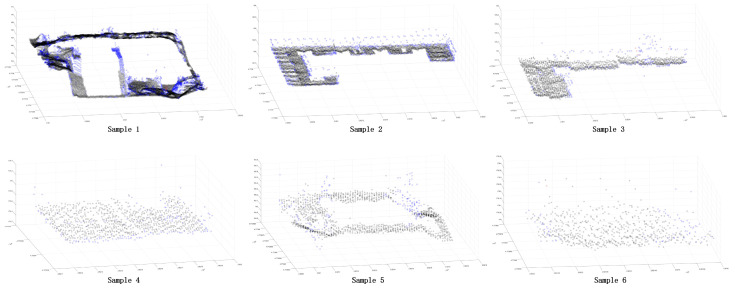
Segmentation error of each sample area using the region-growing method. The black point is the correct segmented point cloud, the red point is the under-segmented point cloud, and the blue point is the over-segmented point cloud.

**Figure 30 sensors-24-00848-f030:**
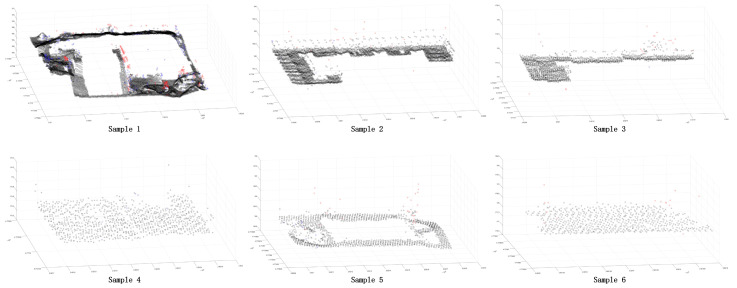
Segmentation error of each sample area using our method. The black point is the correct segmented point cloud, the red point is the under-segmented point cloud, and the blue point is the over-segmented point cloud.

**Figure 31 sensors-24-00848-f031:**
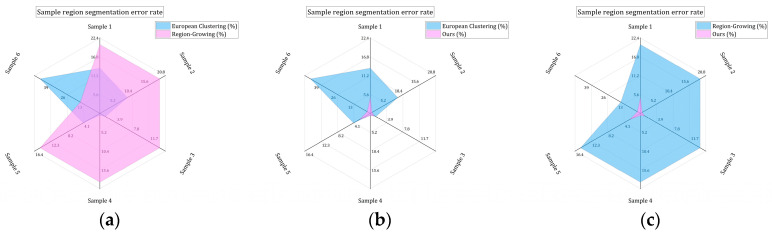
Comparison of error ratios. Panel (**a**) shows a comparison chart of Euclidean clustering and the regional-growing method, panel (**b**) shows a comparison chart of Euclidean clustering and our method, and panel (**c**) shows a comparison chart of the region-growing method and our method.

**Figure 32 sensors-24-00848-f032:**
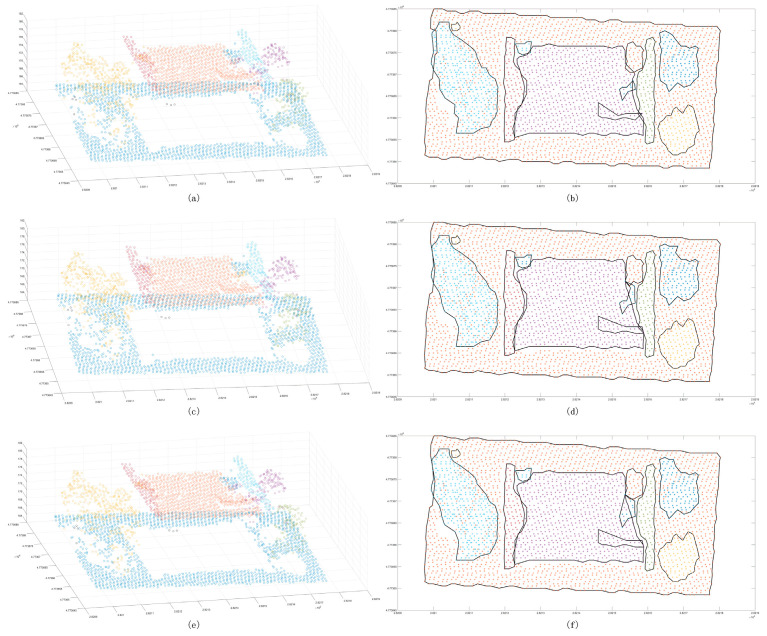
Small-scene segmentation results with different constant coefficients. Panels (**a**,**c**,**e**) show the split scatter plots with constant coefficients of 2, 3, and 4, respectively; panels (**b**,**d**,**f**) show the segmented top view with constant coefficients of 2, 3, and 4. The black lines are the outer contours of different cluster points.

**Figure 33 sensors-24-00848-f033:**
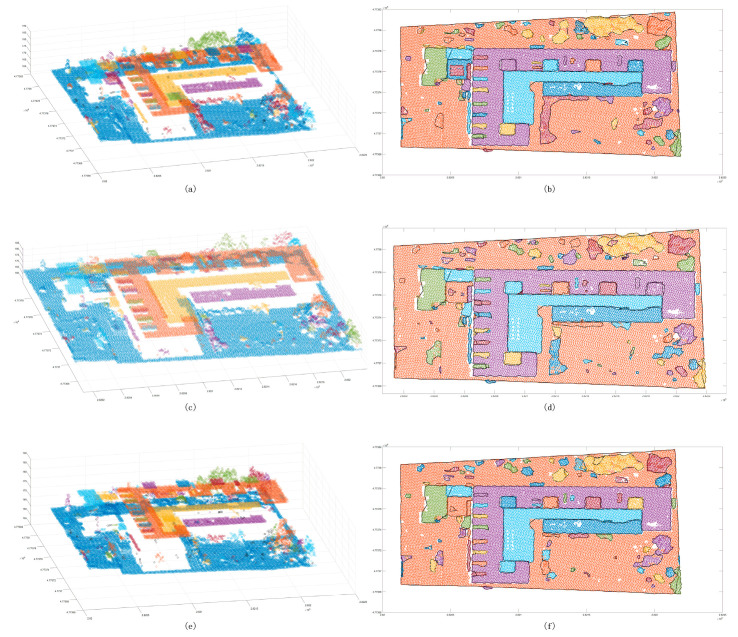
Medium-scene segmentation results with different constant coefficients. Panels (**a**,**c**,**e**) show split scatter plots with constant coefficients of 2, 3, and 4, respectively; panels (**b**,**d**,**f**) show the segmented top view with constant coefficients of 2, 3, and 4. The black lines show the outer contours of different cluster points.

**Figure 34 sensors-24-00848-f034:**
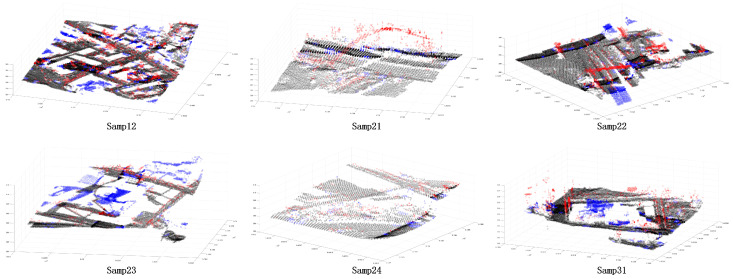
Clustering segmentation results of ground points of different samples. The black points are the correct segmented ground points, the red points are the under-segmented ground points, and the blue points are the over-segmented ground points.

**Table 1 sensors-24-00848-t001:** Experimental data.

Sample	Number of Under-Segmented Points (Red Points)			Number of over-Segmented Points (Blue Points)			Manual Segmentation	Total Error Rate		
	European Clustering	Region-Growing Method	Our Method	Euclidean Clustering	Region-Growing Method	Our Method		Euclidean Clustering	Region-Growing Method	Our Method
Sample1	1753	7	326	179	2943	365	14,410	13.4074	20.4719	4.7953
Sample2	450	0	22	0	981	7	5183	8.6822	18.9273	0.5595
Sample3	28	4	21	0	300	1	2169	1.2909	14.0157	1.0143
Sample4	5	0	0	0	188	6	996	0.5020	18.8755	0.6024
Sample5	47	0	27	14	218	12	1450	4.2069	15.0345	2.6897
Sample6	299	1	15	0	97	0	653	45.7887	15.0077	2.2971
Average								12.3130	17.0554	1.9931

**Table 2 sensors-24-00848-t002:** European clustering segmentation results with a 1.5 m clustering threshold.

Sample	Real Ground Point	Segmentation Point	Accurate Segmentation	Correct Ratio	Under-Segmented Points	Over-Segmented Points
Sample 12	26,691	18,218	16,644	91.36	1569	10,036
Sample 21	10,085	1764	1643	93.14	121	8436
Sample 22	22,504	3088	3014	97.60	74	19,484
Sample 23	13,223	839	835	99.52	3	12,384
Sample 24	5434	960	922	96.04	38	4507
Sample 31	15,556	13,899	12,617	90.78	1276	2933

**Table 3 sensors-24-00848-t003:** European clustering segmentation results with a 2 m clustering threshold.

Sample	Real Ground Point	Segmentation Point	Accurate Segmentation	Correct Ratio	Under-Segmented Points	Over-Segmented Points
Sample 12	26,691	29,825	24,795	83.13	5020	1885
Sample 21	10,085	11,924	10,027	84.09	1891	52
Sample 22	22,504	24,784	20,890	84.29	3888	1608
Sample 23	13,223	11,367	9991	87.89	1372	3228
Sample 24	5434	6357	5377	84.58	975	52
Sample 31	15,556	15,602	13,591	87.11	2005	1959

**Table 4 sensors-24-00848-t004:** European clustering segmentation results with a 2.5 m clustering threshold.

Sample	Real Ground Point	Segmentation Point	Accurate Segmentation	Correct Ratio	Under-Segmented Points	Over-Segmented Points
Sample 12	26,691	35,628	26,453	74.25	9162	227
Sample 21	10,085	12,292	10,079	82.00	2207	0
Sample 22	22,504	25,504	21,352	83.72	4146	1146
Sample 23	13,223	17,985	12,423	69.07	5556	796
Sample 24	5434	6611	5408	81.80	1198	21
Sample 31	15,556	22,120	15,496	70.05	6615	54

**Table 5 sensors-24-00848-t005:** Region-growing segmentation results with a 1.5 m clustering threshold.

Sample	Real Ground Point	Segmentation Point	Accurate Segmentation	Correct Ratio	Under-Segmented Points	Over-Segmented Points
Sample 12	26,691	29,528	20,632	69.87	8884	6048
Sample 21	10,085	9584	8585	89.58	993	1494
Sample 22	22,504	26,353	19,631	74.49	6715	2967
Sample 23	13,223	18,886	11,038	58.45	7844	2181
Sample 24	5434	5091	4249	83.46	837	1180
Sample 31	15,556	19,066	12,952	67.93	6103	2698

**Table 6 sensors-24-00848-t006:** Segmentation results of the combination of RANSAC and the region-growing method [26].

Sample	Real Ground Point	Segmentation Point	Accurate Segmentation	Correct Ratio	Under-Segmented Points	Over-Segmented Points
Sample 12	26,691	20,384	18,467	90.60	1909	8213
Sample 21	10,085	9444	9082	96.17	356	997
Sample 22	22,504	12,198	11,645	95.47	549	10,853
Sample 23	13,223	7224	6313	87.39	908	6906
Sample 24	5434	3067	2877	93.81	186	2552
Sample 31	15,556	13,275	12,218	92.04	1051	3332

**Table 7 sensors-24-00848-t007:** Segmentation results of the method proposed in this paper.

Sample	Real Ground Point	Segmentation Point	Accurate Segmentation	Correct Ratio	Under-Segmented Points	Over-Segmented Points
Sample 12	26,691	25,001	21,695	86.78	3299	4985
Sample 21	10,085	10,272	9015	87.76	1252	1064
Sample 22	22,504	21,989	19,914	90.56	2840	3354
Sample 23	13,223	9816	8799	89.64	1015	4420
Sample 24	5434	5508	4924	89.40	580	505
Sample 31	15,556	13,900	12,438	89.48	1456	3112

**Table 8 sensors-24-00848-t008:** Comparison of removal of non-continuous ground point data.

Sample	Over-Segmented Points	Over-Segmented Points Remove Non-Continuous Ground
Sample 12	4985	2613
Sample 21	1064	1064
Sample 22	3354	1817
Sample 23	4420	1650
Sample 24	505	505
Sample 31	3112	1253

## Data Availability

Data are contained within the article.

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
