# Peer review of "Adaptive Clustering for Point Cloud"

_sensors, 2024, doi:10.3390/s24030848_

Round 1

Reviewer 1 Report

Comments and Suggestions for Authors

The paper presents an adaptive clustering method for segmenting large-scale point cloud data. The clustering thresholds are calculated based on the characteristics of adjacent points. Overall the method shows promise, however, there are several concerns that need addressing:

1\ The paper would benefit from ablation studies to evaluate the impact of key components of the proposed method, such as the factors involved in calculating the adaptive clustering threshold (density, distance, normal vector angle). Analyzing their individual contributions through ablation tests could provide better insights into the importance of each proposed modification over baseline approaches.

2\ There is no explicit discussion of the computational complexity of the proposed method, neither in Method section nor in Experimentation. Understanding the time complexity and memory requirements is essential, especially for large-scale applications.

3\ Although many figures are included for qualitative analysis, the resolution and quality of most figures seems rather low for clear illustrations. Figures meeting a minimum resolution of 300 dpi would better showcase the advantages of the proposed method.

4\ While the paper compares the proposed method with traditional methods, it would be beneficial to see a comparison with current state-of-the-art methods. 

Comments on the Quality of English Language

The paper presents an adaptive clustering method for segmenting large-scale point cloud data. The clustering thresholds are calculated based on the characteristics of adjacent points. Overall the method shows promise, however, there are several concerns that need addressing:

1\ The paper would benefit from ablation studies to evaluate the impact of key components of the proposed method, such as the factors involved in calculating the adaptive clustering threshold (density, distance, normal vector angle). Analyzing their individual contributions through ablation tests could provide better insights into the importance of each proposed modification over baseline approaches.

2\ There is no explicit discussion of the computational complexity of the proposed method, neither in Method section nor in Experimentation. Understanding the time complexity and memory requirements is essential, especially for large-scale applications.

3\ Although many figures are included for qualitative analysis, the resolution and quality of most figures seems rather low for clear illustrations. Figures meeting a minimum resolution of 300 dpi would better showcase the advantages of the proposed method.

4\ While the paper compares the proposed method with traditional methods, it would be beneficial to see a comparison with current state-of-the-art methods. 

Author Response

1.Thank you very much for your advice on our key components, which is indeed an important basis for verifying the method in this paper. However, due to space and time issues, we have included it in the next research plan and used it as an important basis for accurately calculating the adaptive clustering threshold. I believe that the method proposed in this paper can be better improved after following your advice.

2.The practicability of time complexity verification program is indeed a very important index. We add pointer index to the original program, which greatly enhances the operation efficiency, and describe it in the highlight part of the article conclusion.

3.The clarity of the picture is indeed very important to the display of the experimental results. We also uploaded the clear figures in the article, which has a clearer result display.

4.Compared with a variety of methods, the advantages of the proposed method can be more specifically demonstrated. Therefore, we have added a clustering segmentation method combining RANSAC and region-growing, and added multiple research sample areas to further discuss and study the method in this paper.

Reviewer 2 Report

Comments and Suggestions for Authors

In this paper, a segmentation method is presented to cluster LiDAR point cloud. I cannot recommend this paper for publication based on the following comments.

-          Please remove affiliation 1 or 2 after title.

-          There are some huge sentences that are not meaningful. Moreover, it is necessary to edit the manuscript by a native person.

-          Method: there are some workflows in the section. It seems some of them are not necessary or can be presented in one figure.

-          Validation: there are some results regarding boundary and segmentation, but I did not see any numerical validation. Boundary extraction should be validated using ground truth data. Also, segmentation results must be assessed using test samples.

-          According to previous studies, the proposed method is applied in different data sets. Moreover, a comparison between your method and other methods is necessary. Please provide a comparative numeric validation.  

Author Response

1.Thank you very much for your suggestion. We have also revised the title of the article.

2.I am sorry that because our expression has caused trouble to you, we will also use the embellishment service of the publishing house to modify the article after receiving the follow-up article.

3.Thank you very much for your suggestion. For the result verification, we have also added multiple sample areas to verify the article method in more detail, and also generated multiple data comparison tables to reflect the advantages of this method.

4.It is indeed necessary to compare with multiple methods to verify the superiority of the method. Therefore, we added the method proposed this year to compare in the follow-up research process, and generated multiple data comparison tables to further discuss the superiority of the method in this paper.

Reviewer 3 Report

Comments and Suggestions for Authors

The paper proposes an adaptive clustering segmentation method, in which the threshold for clustering points within the point cloud is calculated based on the characteristic parameters of adjacent points, The output data are further refined based on the standard deviation of the cluster points. The test was conducted on different data sets.

The article is well written and is of interest to the readers, however, some changes need to be made:

1.        Figure 3. is not easily understood and is well structured, it is requested to be revised as well as the flow chart in Figure 4.

2.        it is necessary to describe in more detail the processes performed by including more guidance on Graham's Algorithm.

3.        Table 1 should be discussed in more detail.

4.        Discussions need to be better organized and conclusions expanded

Author Response

Thank you very much for your recognition and affirmation of our work. According to your suggestions, we have also modified our work.

1.We have simplified Figure 3 and Figure 4 to facilitate our understanding.

2.We add a pseudo-code settlement process table in the algorithm introduction, and further explain Graham 's Algorithm.

3.We give a more detailed description of Table 1 to further explain the advantages of this method over other methods.

4.We further discuss it in Disscussion, for the language problem, we will also polish it in the follow-up to further explain the method in this paper.

Round 2

Reviewer 2 Report

Comments and Suggestions for Authors

Accept as it is.